# Efficient Supervised Image Clustering Based on Density Division and Graph Neural Networks

**Qingchao Zhao** [1], **Long Li** [1], **Yan Chu** [1,*], **Zhen Yang** [1], **Zhengkui Wang** [2] **and Wen Shan** [3]

1   College of Computer Science and Technology, Harbin Engineering University, Harbin 150001, China
2   ICT Cluster, Singapore Institute of Technology, Singapore 138683, Singapore
3   S R Nathan School of Human Development, Singapore University of Social Sciences,
    Singapore 599494, Singapore
*   Correspondence: chuyan@hrbeu.edu.cn; Tel.: +86-13796675907

**Abstract:** In recent research, supervised image clustering based on Graph Neural Networks (GNN) connectivity prediction has demonstrated considerable improvements over traditional clustering algorithms. However, existing supervised image clustering algorithms are usually time-consuming and limit their applications. In order to infer the connectivity between image instances, they usually created a subgraph for each image instance. Due to the creation and process of a large number of subgraphs as the input of GNN, the computation overheads are enormous. To address the high computation overhead problem in the GNN connectivity prediction, we present a time-efficient and effective GNN-based supervised clustering framework based on density division namely DDC-GNN. DDC-GNN divides all image instances into high-density parts and low-density parts, and only performs GNN subgraph connectivity prediction on the low-density parts, resulting in a significant reduction in redundant calculations. We test two typical models in the GNN connectivity prediction module in the DDC-GNN framework, which are the graph convolutional networks (GCN)-based model and the graph auto-encoder (GAE)-based model. Meanwhile, adaptive subgraphs are generated to ensure sufficient contextual information extraction for low-density parts instead of the fixed-size subgraphs. According to the experiments on different datasets, DDC-GNN achieves higher accuracy and is almost five times quicker than those without the density division strategy.

**Keywords:** supervised clustering; face clustering graph neural network; link prediction

## 1. Introduction

Due to the prevalence of cameras, a lot of photographs are created every day in the modern world (e.g., smartphones, monitor systems, etc.). Unfortunately, this has created significant difficulties for image management systems [1–4]. For instance, to better manage face images, face clustering has emerged as a highly popular method to handle face photos for tasks such as face recognition, face labeling, etc. [5–8].

Conventional clustering techniques (e.g., K-means [9], DBSCAN [10], HAC [11], Spectral Clustering [12], etc.) rely on certain assumptions and are hard to handle face images with complicated data distributions. For example, K-Means requires the datasets to be convex-shaped, and DBSCAN requires different clusters with similar densities. Supervised clustering techniques have been developed to learn cluster patterns based on Graph Neural Networks (GNNs) link prediction [13] and have demonstrated significant improvements over conventional approaches in terms of accuracy [14,15]. These link prediction-based algorithms can learn the real similarity between instances and have less strict constraints for the data distribution which can enhance the effectiveness of face clustering. These kinds of methods share similar operations. The first is the construction of subgraphs for each instance as the central node first, followed by the prediction of the linkage probability between the central node and its neighbors. In the end, all instances are grouped into

clusters according to the predicted connectivity. However, these methods have produced large computational overheads despite being sufficient to identify more precise clusters. The main computation of this type of method is a large number of subgraphs generations for each image instance and inferring the similarity between the instance and its neighbor. A basic observation is that a central node can be a neighbor node in other subgraphs and the produced subgraphs are usually highly overlapped. This increases the cost of duplicated calculations excessively and slows down the clustering speed.

To resolve the shortcomings in supervised clustering, we use density division clustering to improve the supervised clustering and propose the GNN-based supervised clustering framework DDC-GNN, which is both time-efficient and effective. In the first step, density division is performed to divide the image instances (or referred to as nodes) into high-density parts and low-density parts based on their density. The high-density parts are the data instances which locate at the cluster center. Then a subgraph is generated for each node in low-density parts, and a GNN model is used to perform link prediction for each subgraph. Finally, the pseudo-label propagation approach merges the nodes in low-density parts and high-density parts by a predicted linkage similarity. After several iterations in which the unqualified edges are deleted, the final clusters are formed.

A preliminary version of this paper has been published in [16]. In this paper, we have incorporated numerous extensions, including a general framework of GNN-based supervised clustering based on density division, evaluating two different types of GNN models (i.e., the GCN-based model and the GAE-based model) under the framework, a more comprehensive experiment evaluation and analysis together with numerous expansion information thorough the paper. In particular, we leverage two different types of GNN models in the connectivity learning and prediction phase, which are the GCN-based model and the GAE-based model. The difference between the two models is obvious. The GCN-based model deals with a node classification task to neighbor nodes. A neighbor node may have the same label as the center node or have a different label from the center node. The encoding manner of the center node feature in GCN is to build the edge between the center node and the neighbor node, which means that the feature of the center node is encoded in the GCN model by message passing/propagation. While the GAE-based model is a link prediction/classification task. A linkage probability between the center node and neighbor node will be predicted to be from 0 to 1, which is the output of the decoders in the GAE-based model. It is the multiplication of the hidden center node feature and the hidden neighbor node feature. The encoding manner of the center node feature in the GAE-based model is the inner product of the hidden feature.

The contributions of our paper are summarized as follows:

- We propose a novel time-efficient and effective GNN-based supervised clustering framework based on density division DDC-GNN. The main phrases are the density division phase, the GNN model inferring phase, and the cluster requiring phase. In GNN model inferring phase, any Plug-and-Play model for image similarity inferring can be applied. In this paper, we have tested GCN-based and GAE-based models.
- To reduce the high computation overhead in existing supervised image clustering algorithms, we present a novel density division method to split the face image instances into high-density parts and low-density parts, where only the low-density parts of the nodes need to perform the connectivity prediction. The density division strategy drastically reduces the computing overhead and unnecessary inferring.
- Extensive experiments are implemented to compare the DDC-GNN framework with other approaches using the MS-Celeb-1M and IJB-B face datasets. The results demonstrate that DDC-GNN is five times quicker than existing approaches with comparable accuracy in a fair comparison.

This paper is organized as follows. Section 2 introduces the related work about the graph neural network and face clustering. Section 3 shows the details of the proposed DDC-GNN framework. Sections 4–6 provide the experimental evaluations, discussion, conclusion and future work respectively.

## 2. Related Work

### 2.1. Graph Neural Networks

In general, convolutional neural networks (eg. Mask R-CNN [17], etc.) cannot process non-grid spatial structure data while graph convolutional networks can solve this problem perfectly. Andriyanov et al. were to develop an algorithm that could help increase segmentation accuracy when used together with the known algorithms [18]. In semi-supervised node classification tasks, Kipf et al. [19] proposed the graph convolutional networks, which significantly enhanced the performance of node classification. Schlichtkrull et al. [20] solved link prediction and entity classification in knowledge graph mapping tasks by graph convolutional networks. Liu et al. [21] applied graph convolutional networks to capture the global information to solve text classification. Graph auto-encoder also used GCN layers to predict linkage [22]. GraphSAGE extended the graph neural network to inductive learning by fusing the neighbor nodes of the current node to obtain the feature representation [23]. Each neighbor node is treated equally in GraphSAGE. But in actual scenarios, different neighbor nodes may play different roles. Attention was a good mechanism and widely used [24]. The Graph Attention Network (GAT) was introduced to aggregate the neighbor nodes through the self-attention mechanic, which realized the adaptive matching of the weights of different neighbors, thereby improving the accuracy of the model [25]. In this paper, we leverage the powerful information aggregation of the graph neural network to learn the potential relationship between nodes and infer connectivity between nodes.

### 2.2. Unsupervised Face Clustering

Face clustering is a typical task that organizes images of faces into groups according to their identities, ensuring that faces in one group are all from the same identity and faces in separate groups have different identities. It has received much attention as a basic data mining learning task. The conventional clustering techniques, such as K-means [9], DBSCAN [10], mean shift [26], rely on certain assumptions, making it challenging to cluster complicated face images because of high dimension features, face position, light, and other factors. To organize face image instances into clusters, Lin et al. [27] presented a hierarchical clustering algorithm based on linear support vectors. In order to handle data with an uneven distribution of density, a density-aware cluster-level affinity measure that used singular value decomposition was also created [28]. Zhu et al. [29] converted the face clustering problem into a multi-image cutting problem by the gradient flow method. For face clustering, Tapasw et al. [30] suggested a ball clustering learning method that could automatically count the number of clusters in the video.

### 2.3. Supervised Face Clustering

Supervised learning has been used for face clustering in a recent study. Zhan et al. [31] tried to bring input data into several models for each pair of nodes and including numerous commentator networks and then trained a multi-layer perception classifier to gather the information. Wang et al. [2] combined the subgraphs information using the GCN's aggregation capabilities, which significantly enhanced the prediction of inter-node linkages. Yang et al. [14] developed an approach to produce multi-scale subgraphs as input for a graph convolutional neural network to detect cluster proposals. Zhao et al. [32] analyzed data distribution with supervised learning and used an effective framework to cluster new data in an incremental context. Qi et al. [33] used ResNet-GCNs to discover the connectivity between nodes. Yang et al. [15] employed a Reverse-Imbalance Weighted Sampling (RIWS) technique to improve graph representations in unbalanced labeled environments.

Despite the fact that supervised face clustering has been shown to be more efficient, it has a significant computational cost because they all need to produce a large number of subgraphs for connection learning. Moreover, these subgraphs are usually overlapped. For instance, L-GCN [2] needs a lot of subgraphs to detect the cluster pattern (linkage between nodes) properly. There are many neighbors between each subgraph, especially close to the cluster center, which are overlapping. However, our proposed framework focuses on

predicting the connection of the low-density part after dividing face images by the density. Since the face image instances from the high-density parts are naturally connected in the cluster center, fewer computing resources are required and the clustering proceeds more quickly and accurately.

## 3. Supervised Clustering Based on Density Division

### 3.1. Overview

To better appreciate the proposed solution, we first provide a framework overview of DDC-GNN. The DDC-GNN framework overview is presented in Figure 1, which consists of three main stages.

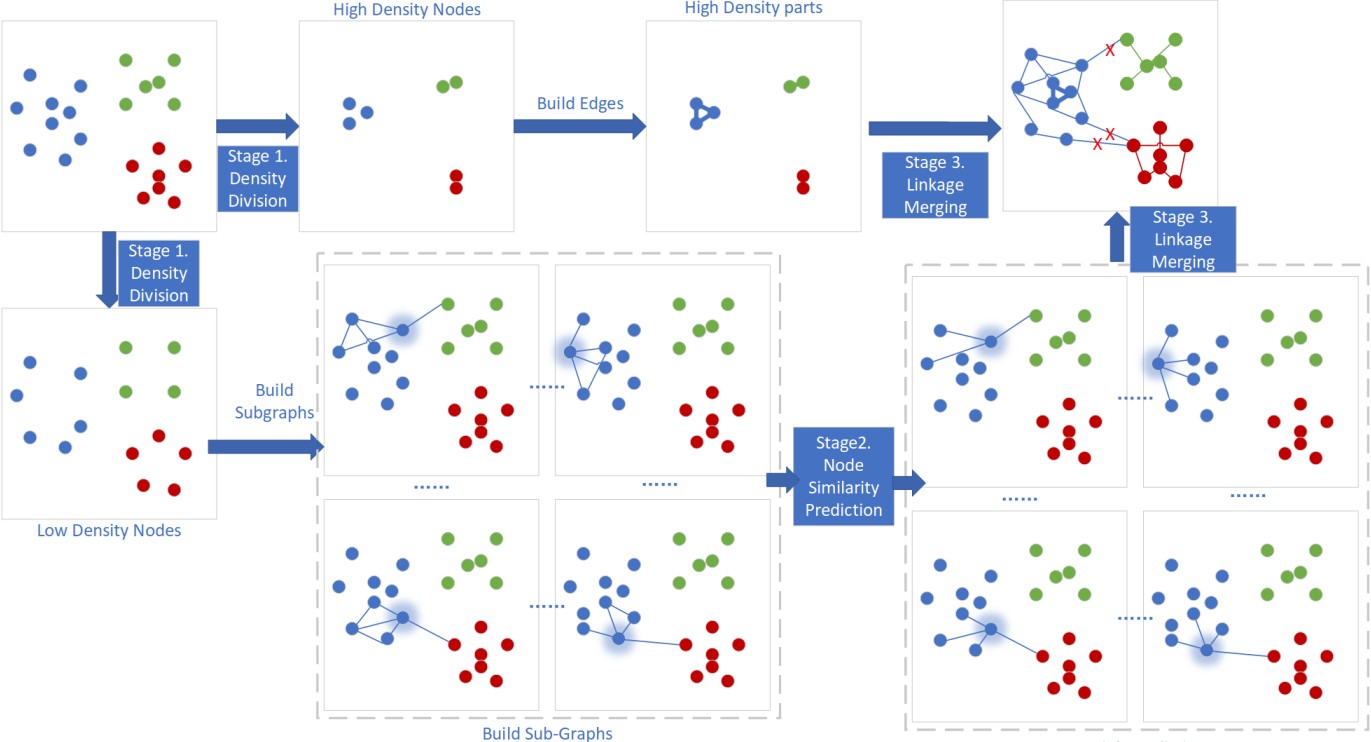

**Figure 1.** The framework of DDC-GNN. Firstly, the face data are divided into high-density and low-density parts. The high-density parts are located at the center of the cluster and have strong connectivity. Secondly, we use each data point of the low-density parts as a node, and construct an ASG for each node only in low-density parts, then apply GNN on ASGs to predict the connectivities among the nodes. After that, all the nodes are connected into a big graph with their linkage and clusters are obtained by iteratively deleting unnecessary links.

The first stage of the framework is the node division by density. In general supervised clustering algorithms, most of the approaches need to predict the connectivity/linkage of edges between all the nodes. Unlike the previous work, the DDC-GNN framework does not need to predict the linkage for the entire face image sets. Instead, DDC-GNN divides the entire face image sets into high-density parts and low-density parts. The nodes in the high-density parts are close to the center of the cluster, which tends to be in the same cluster with strong connectivity and do not need to make predictions.

In the second stage of the framework, for each node from the low-density parts, a customized adaptive subgraph (ASG) is built and learned by a GNN model (GCN or GAE model) for linkage/connectivity prediction. These ASGs can be used as an input by the GNN model to predict connectivity that reflects real image instance similarity.

The third stage of the framework is linkage merging to form the clusters. A pseudo label propagation approach organizes the nodes from high-density and low-density parts into clusters.

### 3.2. Notations and Descriptions

The notations we used are shown in Table 1. The formal expression of our image clustering task is $C = DDC\text{-}GNN(V)$, where $C$ is the cluster results and $V$ is the node-set with extracted feature $F$ from a pre-trained model.

**Table 1.** Notations and Descriptions.

| Notations | Descriptions |
|---|---|
| $V, v$ | $V$ is a set of nodes in the graph, and $v$ is an instance that belongs to $V$. |
| $A$ | $A$ is the adjacency matrix of a graph. |
| $F$ | $F$ is the feature set of extracted features from the pre-trained model. |
| $C$ | $C$ is the cluster result. |
| $D$ | $D$ is the dimension of the features. |
| $N$ | $N$ is the number of face images. |
| $\Lambda$ | $\Lambda$ is a diagonal matrix. |
| $d$ | $d$ is a set of node density values. |
| $W$ | $W$ is a learnable weight matrix. |
| $Z$ | $Z$ is the hidden layer feature. |
| $Vs$ | $Vs$ is the node set of the subgraph. |
| $Es$ | $Es$ is the edge set of the subgraph. |
| $G(Vs, Es)$ | $G(Vs, Es)$ is the input subgraph of GNN. |
| $F_p$ | $F_p$ is the feature set of the subgraph which is built for node $p$. |

### 3.3. Connected Region Division

As the existing linkage-based face clustering approaches need to form subgraphs for each face (node), there are a large number of subgraphs generated that result in high computation overhead. To reduce the overhead, our strategy is to divide the nodes into high-density parts and low-density parts. The high-density parts refer to the nodes which are strongly connected with each other and likely in the same cluster in the end. Otherwise, the nodes belong to the low-density parts which need to predict their connectivities using the GNN.

To do so, we first calculate the distance between the nodes. The distance can be easily calculated based on the existing measures like cosine similarity and so on. Then, we calculate the density of each node based on their distances to other nodes. We define the density of node $v$ as the number of nodes that are within $E$ distance, where $E$ is a predefined distance threshold for density calculation. After calculating the distance, we divide the nodes into high-density parts and low-density parts. For instance, we can define the proportion $\rho$ of entire nodes as the high-density nodes, where $\rho$ can be obtained from the training dataset. Next, among all the high-density nodes, we further form them into high-density components by linking those high-density nodes. For any given high-density nodes $m$ and $n$, if $m$ is one of the $M$ nearest nodes of $n$, we link $m$ and $n$ with one edge, where $M$ is another parameter obtained from the dataset. In other words, if the distance between $m$ and $n$ is smaller than other $M$ nodes with $n$, $m$ will not be linked to $n$. This is to make sure that the high-density nodes linked together are strongly connected to each other. Thus, they can be safely linked together as a strongly connected component. Those nodes in a strongly connected component are likely in the same cluster. Thus, the edge connectivity weights between these nodes in the same high-density component can be set as 1, as they will be in the same cluster.

By so doing, the high-density nodes are grouped into various strongly connected components with assigned weights.

For example, we set $E = 0.5$, $\rho = 50\%$, $M = 2$ and there are four nodes. The distances between nodes are shown in the following matrix.

$$\begin{pmatrix}
 & node\ 1 & node\ 2 & node\ 3 & node\ 4 \\
node\ 1 & 0 & 0.1 & 0.3 & 0.5 \\
node\ 2 & 0.1 & 0 & 0.8 & 0.4 \\
node\ 3 & 0.3 & 0.8 & 0 & 0.6 \\
node\ 4 & 0.5 & 0.4 & 0.6 & 0
\end{pmatrix} \tag{1}$$

The densities of four nodes are 2,2,1,1 and $d = \{2, 2, 1, 1\}$. When we choose *node* 1 and *node* 2 are two high-density nodes and *node* 3 and *node* 4 are low-density nodes.

### 3.4. Node Similarity Prediction of Adaptive Subgraph in GNN

The calculation of similarity/connectivity/linkage between nodes is usually based on non-learnable Euclidean distance or non-learnable cosine similarity. In recent works, supervised similarity learning has been applied to GCN and obtains promising results, such as the node classification model in L-GCN [2] and GCN-VE. Our paper summarizes the mainstream methods into a unified connectivity prediction framework, namely: 1. Locate nodes of subgraphs; 2. Build edges of subgraphs; 3. Apply a GNN model on ASGs. Under the connectivity prediction framework, we generate adaptive subgraphs instead of previously fixed subgraphs and design two types of models (i.e., node classification model and link prediction model). Based on the method, when a GCN node classification model is used for connectivity prediction in the framework, it is named DDC-GCN. When a GAE link prediction model is used for connectivity prediction, we call it DDC-GAE.

#### 3.4.1. Node Classification Model for Connectivity Prediction

According to the neighborhood context information of two face images (nodes), their connectivity can be learned by ASGs, which are constructed to extract the real similarity information of neighboring nodes. The ASGs generation is shown in Figure 2.

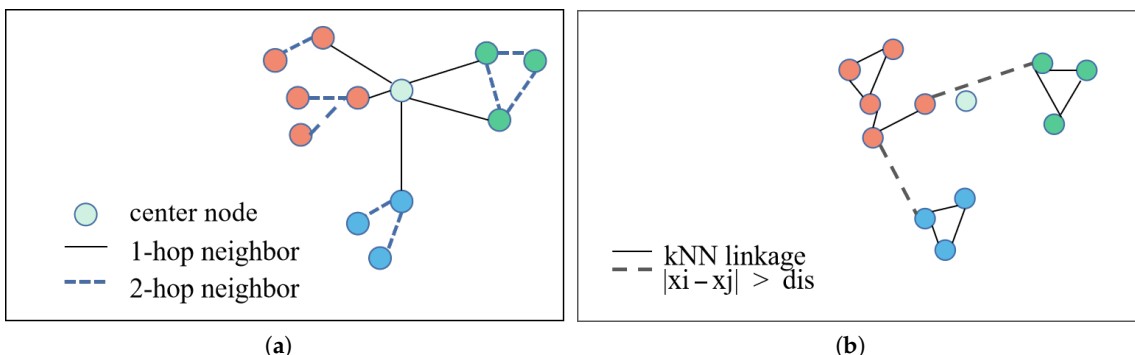

**Figure 2.** Construction of adaptive subgraph. (**a**) Finding nodes; (**b**) Adding edges.

The first phase is locating nodes in a certain ASG. For a given node as the central node $p$, we locate second-hop neighbor nodes of $p$ to form an ASG. We denote ASGs as $G(Vs, Es)$. For each hop of neighbor nodes, we construct a subgraph by locating $h_1$ one-hop direct neighbors and $h_2$ two-hop indirect neighbors of each central node. $h_1$ one-hop neighbors are the $h_1$ nearest neighbors of $p$ and $h_2$ two-hop neighbors are the $h_2$ nearest neighbors of one-hop neighbors.

The second phase is building edges for all nodes in a certain ASG. We calculate an adaptive average distance $dis$, iteratively obtain the maximum Euclidean distance between each direct neighbor (one-hop) and its $h_2$ indirect neighbors (two-hop), and then calculate the average of these maximum values to obtain the final adaptive distance. For a node $q \in Vs$, we look for potential edges in the $k$ nearest neighbors of $q$. If a node $r$ appears in $Vs$ and the distance between $q$ and $r$ is less than $dis$, the edge $(q, r)$ is added to the set $Es$. This

process can ensure that sufficient structure of the subgraph. Then we get the subgraphs with central node $p$, the node-set $Vs$, and the feature set $F_p$.

$$F_p = [\dots, f_s, \dots]^T, s \in V_s \quad (2)$$

Finally, we finish the construction of the adaptive subgraph, obtaining the characteristic matrix $F_p$ and adjacency matrix $A_s \in R^{(|V_s| x |V_s|)}$.

The context information in ASGs can be used to predict the connectivities of low-density data instances. A four-layer GCN model is used to perform node classification tasks on ASGs which reflects the similarities between nodes and the formal expression of each layer is provided as follows.

$$F_{(l+1)} = \sigma([F_l \parallel GF_l]W) \quad (3)$$

where $F_l \in R^{(N \times D_{in})}$, $F_{(L+1)} \in R^{(N \times D_{out})}$, $N$ is the number of nodes, $D_{in}$ is the input dimension of the node feature, $D_{out}$ is the output dimension of the node feature, $F_l$ is the output characteristic of the $l$ layer, $W$ is a learnable weight matrix for the convolution layer, which has a size of $2d_{in} \times d_{out}$. $G = \Lambda^{(-1/2)} A \Lambda^{(-1/2)}$ is an aggregation matrix of size $N \times N$, each row of which adds up to 1, where $A$ is the adjacency matrix and $\Lambda$ is a diagonal matrix. The diagonal element is $\Lambda_{II} = \Sigma_j A_{ij}$. The operator $\parallel$ means that the matrix is joined along with the feature dimension. $\sigma(\cdot)$ is a nonlinear activation function. There are two steps in the graph convolution process. In the first step, the potential neighbor information of the node is gathered by the operation that $F_l$ multiply $G$, and then the input node features $F_l$ and $G \cdot F_l$ are concatenated along the feature dimension. In the second step, the feature matrix is transformed into the input of the next layer $F_{(L+1)}$ through a set of learnable parameters $W$.

As shown in Figure 3, A is the adjacency matrix and X is the feature matrix. The node classification model GCN of DDC-GNN is a stack of four convolutional layers of graphs, and the *ReLU* nonlinear activation function of each GCN layer is used for activation. To predict the existence of edges between nodes in subgraphs, we use a two-layer MLP classifier to minimize the cross-entropy loss between the predicted connectivity and the ground-truth edge label. We use a pair of node features corresponding to each edge in the subgraph as the input of MLP to obtain the two-dimensional connectivity.

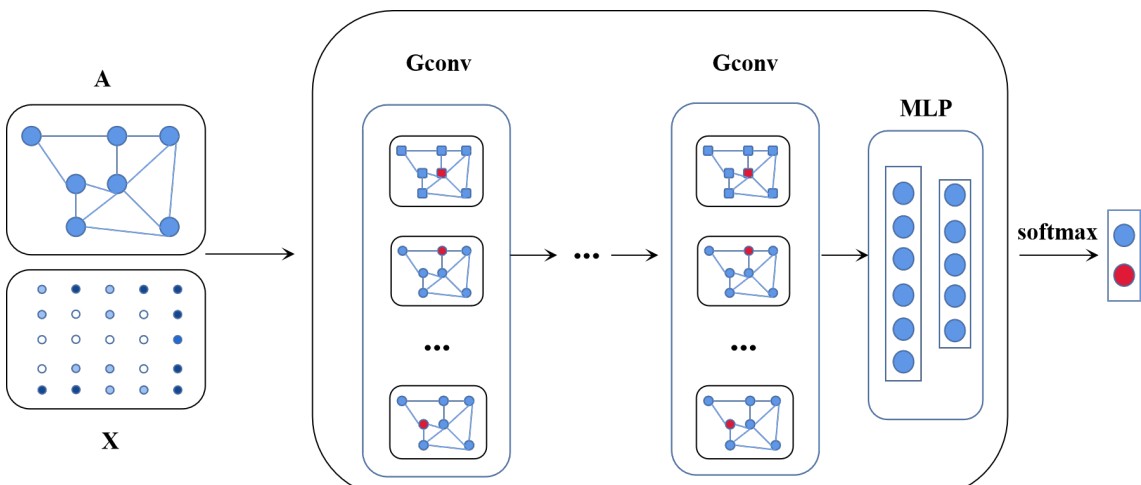

**Figure 3.** The structure of node classification model GCN.

In the training phase, for a neighbor node that is in the same cluster as the central node, the connectivity between this neighbor and the central node is set to 1 (with the same label) and the connectivity is set to 0 when the neighbor node is in a different cluster with the central node (with a different label). For the inferring phase, the output of GCN is from

0 to 1 built by the softmax function. The architecture of the node classification model in Table 2.

**Table 2.** The architecture of the node classification model GCN.

| Method | Layers |
|:---:|:---:|
| GCN-based | 4 Layers GCN<br>2 layers MLP<br>Softmax Activate Function<br>cross-entropy Loss Function |

The cross-entropy loss function is defined as the node-wise cross-entropy loss overall labeled instances.

$$cross\text{-}entropy\ loss = \sum_{v\in V}\sum_{C_{dim}=1}^{C_{dim}} L(v)[f]\cdot \log F_{hidden}[f] \tag{4}$$

$V$ is a set of nodes in the graph, and $v$ is a node belonging to $V$. $C_{dim}$ is the output dimension, which is equal to the number of classes. There are positive instances and negative ones in our node classification model. Positive instances are the neighbor nodes with the same label as certain center nodes and negative instances are the neighbor nodes with different labels, so $C_{dim}$ is 2 in the node classification model. Given an input training set in the model, we can set a two-dimensional vector label by the label indicator matrix $L(v)$.

$$L(v) = \begin{cases} (1,0), & l_v = l_{center\ node} \\ (0,1), & l_v \neq l_{center\ node} \end{cases} \tag{5}$$

When the label of a neighbor node is in the same class as the center node, the label vector is (1, 0). While a neighbor node is in a different class from the center node, the label vector is (0, 1).

After we divide the data set, we construct an adaptive subgraph for each node in the low-density nodes and predict the possibility of a connection between each node in the subgraph. Therefore, we get a set of edges with the predicted connectivity which indicates the similarity between a pair of nodes.

### 3.4.2. Link Prediction Model for Connectivity Prediction

The node classification task predicts the probability that the neighbor node shares the same class label with the central node. Different from the node classification model, the link prediction model is to predict the edge connectivity probability between two nodes in a network. Graph auto-encoder is a typical model which is used to perform link prediction tasks. The ASGs generation in our designed link prediction model is the same as the node classification model, while the supervised information is different. In a node classification task, a one-hop neighbor node has a label with 0 or 1. Differently, an edge is assigned a label with 1. If two nodes (the central node and its one-hop neighbor node) have no edge, the position of the adjacent matrix is set to be 0.

As the construction of the link prediction model is shown in Figure 4, the GAE model has an encoder and decoder structure. The encoder has $n$ GCN-layers and the decoder is a matrix multiplied to reconstruct the adjacent matrix.

The formal expression of the encoder is as follows.

$$Z = GCNlayer(X, A) \tag{6}$$

The formal expression of the decoder is:

$$\hat{A} = \sigma(ZZ^T) \tag{7}$$

where $GCNlayer(X, A) = \tilde{A}ReLU(\tilde{A}XW_0)W_1$ and $\tilde{A} = \Lambda^{(-1/2)}A\Lambda^{(-1/2)}$. $\Lambda$ is a diagonal matrix. $Z$ is the hidden variable of the hidden layer and $X$ is the feature matrix and $A$ is the adjacent matrix. $\hat{A}$ is the reconstruction adjacent matrix. We minimize the cross-entropy reconstruction loss of $\hat{A}$ and $A$. The architecture of the link prediction model is shown in Table 3.

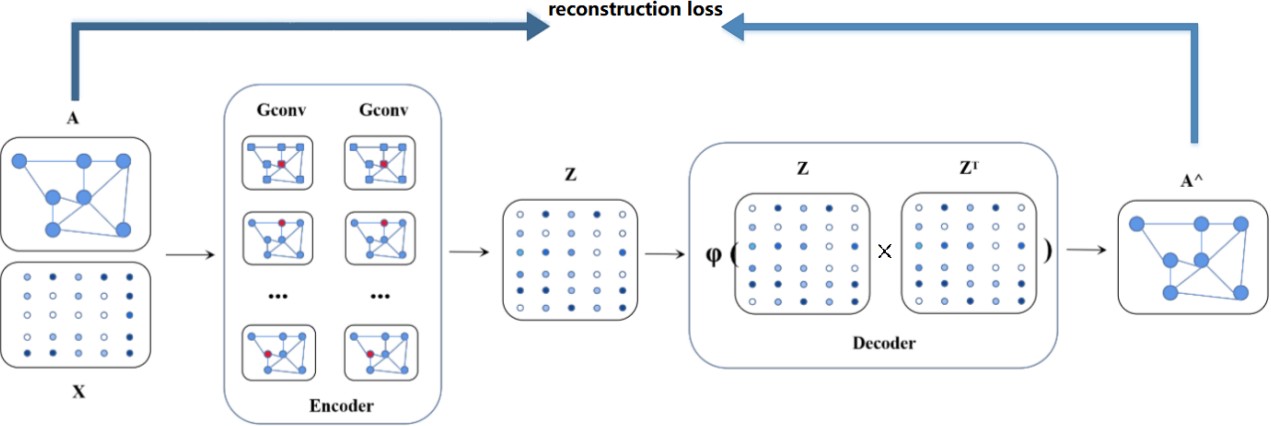

**Figure 4.** The structure of link prediction model GAE.

**Table 3.** The architecture of the link prediction model GAE.

| Method | Layers |
|:---:|:---:|
| GAE-based | Encoder: 2 Layers GCN |
| | Decoder: Inner Product |
| | Sigmoid Activate Function |
| | cross-entropy Loss Function |

In the training phase of the link prediction model GAE, as a neighbor node is in the same cluster with the central node, the connectivity between this neighbor and the central node is set to 1 (with the same label) and the connectivity is set to 0 when the neighbor node is in a different cluster with the central node (with a different label).

The loss function is defined as the node-wise cross-entropy loss overall labeled instances.

$$cross\text{-}entropy\ loss = \sum_{i,j \in V} \sum_{C_{dim}=1}^{C_{dim}} L(i,j)[f] \cdot \mathbf{log}\hat{A}_{i,j}[f] \tag{8}$$

where $(i, j)$ is the set of node pairs in the graph belonging to $V$. $C_{dim}$ is the output dimension of the linkage, which is equal to the number of classes. There are two classes in our graph auto-encoder model, which is whether there should be a linkage between the node pairs, so $C_{dim}$ is 2 in the GAE model. We can set a two-dimensional vector label by the label indicator matrix $L(v)$ when we get the input training set into the model.

$$L(v) = \begin{cases} (1,0), & connectivity = 1 \\ (0,1), & connectivity = 0 \end{cases} \tag{9}$$

When there should be a linkage and the connectivity is 1, the label vector in the indicator matrix is (1, 0). While connectivity between two nodes is 0, the label vector in the indicator matrix is (0, 1).

### 3.5. Linkage Merging

To obtain clusters, a simple method is to set a threshold, cut out all edges with less weight than the threshold, and propagate the pseudo-labels using Breadth First Search.

However, this method is sensitive to the threshold value. Therefore, we adopt the pseudo-label propagation strategy proposed in [31]. In each iteration, edges with weights less than the threshold are cut off, and the remaining edge instances are connected to form clusters. If a cluster in these clusters is larger than the predefined maximum size of the cluster, it will be added to the queue for processing in the next iteration. The threshold which is used to cut edges increases gradually during iteration, and then stops iterating when the queue is empty, which means the clustering is complete.

*3.6. Efficiency Analysis*

Three components of the algorithm-density division, GNN connectivity prediction, and link merging-contribute mostly to its complexity. Approximate Nearest Neighbor Search ($O(nlogn)$) is more efficient than the brutal search ($O(n \times n)$) in neighbor node locating. The density value is calculated linearly in the density division. The complexity of the density division is $O(n \times M)$ because $M$ nearest neighbor nodes of each node with high density must be found to create the edge in the high-density parts. The creation of ASGs in the low-density parts is the main component of connectivity prediction. The complexity of $O(nlogn)$ can be much lower than the ANN search in the ASG creation process. Link merging requires fewer computations and a lower complexity $O(n \times h1)$. Compared with $n$, $M$ and $h_1$ are two small number. As a result, the time complexity of DDC-overall GNN is $O(nlogn)$. The DDC-GNN separates the data into two portions with differing densities and only applies GCN or GAE inference to the low-density parts. Because the computation required for ASGs inferring reduces greatly, clustering of the DDC-GNN framework is more effective.

Algorithm 1 is the proposed supervised clustering based on the density division algorithm.

---

**Algorithm 1** supervised clustering based on density division

---

**Input:** the number of neighbors $M$, feature set $F$, distance threshold $E$, descending density set of all nodes $d$, $model\_type = GAE/GCN$

**Output:** Clusters $C$
1: **procedure** DDC-GNN
2:     Edge, $D_l$ = Density Partition($E, M, d$)
3:     Edge = Edge $\cup$ Connectivity Prediction(F, $D_l$, $model\_type$)
4:     $C$ = Use pseudo-label propagation strategy proposed in [31] to find connected graphs on Edge
5:     **return** $C$
6:
7:     **function** DENSITY PARTITION($E, M, d$)
8:         Divide $d$ into two sets according to the Ratio: high-density $D_h$ and low-density $D_l$
9:         **for** $i = 1, 2, 3, \ldots n$ in $D_h$ **do**
10:           Find $M$ nearest neighbors $v_i \in N(v_i)$
11:           **if** dis($v_i, v_j$) < $E$ and $v_j \in N(v_i)$ **then**
12:             Edge = Edge $\cup$ Edge$\{v_i, v_j\}$
13:         **return** Edge, $D_l$
14:
15:     **function** SIMILARITY PREDICTION(F, $D_l$, $model\_type$)
16:         **if** $modeltype == GAE$ **then**
17:           Connectivity=GAE(F, $D_l$)
18:         **if** $modeltype == GCN$ **then**
19:           Connectivity= GCN(F, $D_l$)
20:         **return** Connectivity

---

## 4. Experiments and Evaluations

### 4.1. Experiments Setting

#### 4.1.1. Dataset

DDC-GNN is evaluated by the publicly available benchmarks for large-scale face recognition, the MS-Celeb-1M [34] and IJB-B [35]. In total, there are 100k identities that constitute MS-Celeb-1M. The number of images varies from 10 to 600 in one identity in the CT images [36] and the remote sensing images [37]. In order to divide them into a training set and a testing set, we apply the same parameters as in [14]. The numbers of identities included in the three subtasks are 512, 1024, and 1845, respectively. And the numbers of samples included are 18,171, 36,575, and 68,195, respectively. We use the model trained on a random subset of CASIA [38] datasets for testing clustering.

#### 4.1.2. Metrics

In order to evaluate the performance of the proposed face clustering method in a comparable way, we use the most common evaluation metrics in image clustering task in the same experimental environment: B-cubed F-score (denoted as *F*), Normalized Mutual Information (denoted as NMI) [39], Adjusted Rand Index (denoted as ARI) and Silhouette Coefficient (denoted as SC).

***NMI*:** Given the true label $\Omega$ of a cluster, $C$ is the predicted cluster label, and the definition of NMI is as follows:

$$NMI(\Omega, C) = \frac{I(\Omega, C)}{\sqrt{H(\Omega)H(C)}} \tag{10}$$

$$I(\Omega, C) = H(\Omega) + H(C) - H(\Omega, C) \tag{11}$$

$$H(\Omega) = -\sum_{\omega} P_{\Omega}(\omega) \log P_{\Omega}(\omega) \tag{12}$$

$$H(C) = -\sum_{c} P_C(c) \log P_C(c) \tag{13}$$

$$H(\Omega, C) = -\sum_{\Omega, C} P_{\Omega, C}(\omega, c) \log P_{\Omega, C}(\omega, c) \tag{14}$$

Among them, $H(\cdot)$ is the entropy function, respectively, and $I(\Omega, C)$ represents mutual information. $P_{\Omega}(\omega)$ and $P_C(c)$ are the distribution of $\Omega$ and $C$. $P_{\Omega, C}(\omega, c)$ is the joint distribution of $\Omega$ and $C$.

***F-score*:** *F-score* (denoted as *F* in the following paper) is the harmonic average of precision and recall. $C(i)$ and $L(i)$ are to represent the cluster label and true label of node $i$, respectively. And we define the accuracy of two nodes $i$ and $j$ as follows.

$$\text{Correctness}\,(i,j) = \begin{cases} 1, & \text{if } L(i) = L(j) \text{ and } C(i) = C(j) \\ 0, & \text{if otherwise} \end{cases} \tag{15}$$

The definitions of precision and recall are as follows.

$$\text{Precision} = \frac{1}{N} \sum_{i=1}^{N} \sum_{j \in C(i)} \frac{\text{Correctness}(i,j)}{|C(i)|} \tag{16}$$

$$\text{Recall} = \frac{1}{N} \sum_{i=1}^{N} \sum_{j \in L(i)} \frac{\text{Correctness}(i,j)}{|L(i)|} \tag{17}$$

where $|C(i)|$ and $|L(i)|$ represent the size of sets $C(i)$ and $L(i)$, respectively. The definition of *F-score* is as follows.

$$BCubed\_F - score = \frac{2 \times \text{Precision} \times \text{Recall}}{\text{Precision} + \text{Recall}} \tag{18}$$

***ARI:*** *ARI* measures the coincidence degree of two data distributions. The higher the value is, the more the clustering result matches the real situation. $ARI \in [-1, 1]$, and it can be expressed as:

$$ARI = \frac{RI - E(RI)}{max(RI) - E(RI)} \tag{19}$$

where *RI* is Rand Index which:

$$RI = \frac{a + d}{C_N^2} \tag{20}$$

$C_N^2$ is the data pair in all data and $N$ is the total sample number. $a$ is the logarithm of data pairs in the same category in true clusters and also in the same category in predicted clusters at the same time. $d$ is the logarithm of data pairs not in the same category in true clusters and not in the same category in predicted clusters.

***SC:*** $s(i)$ of a sample $i$ can be expressed as follows:

$$s(i) = \begin{cases} 1 - \frac{a(i)}{b(i)} & , a(i) < b(i) \\ 0 & , a(i) = b(i) \\ \frac{a(i)}{b(i)} - 1 & , a(i) > b(i) \end{cases} \tag{21}$$

where $a(i)$ is the average distance from sample $i$ to other samples in the same cluster and $b(i)$ is the average distance from sample $i$ to all samples of other clusters. Silhouette coefficient of sample $i$ is defined according to the intra-cluster dissimilarity $a(i)$ and inter-cluster dissimilarity $b(i)$ of sample $i$. The silhouette coefficient of the predicted cluster is the average silhouette coefficient of all samples.

### 4.1.3. Parameter Setting

We investigate several values for the parameters $M$ and $E$ and discover that the clustering effect is best when $M$ is set to the number around 0.1 percent of the data volume and the distance $E$ for cosine distance is set to 0.8. In addition, several experiments lead to the density selection threshold being at 0.4. The weight decay value is $1 \times 10^{-5}$ and the initial learning rate is 0.1.

### 4.1.4. Benchmarking

Our method is compared with these clustering methods, which are described briefly as follows.

- K-Means [9]: K-Means divides a data set into several clusters according to how far between data instances. The cluster's sample distances are reduced as much as possible via K-Means.
- Affinity Propagation (AP) [40]: All data points should be treated as potential clustering centers, and the information transmission between the data instances is taken into consideration during clustering, which is appropriate for high-dimensional, multi-class data clustering algorithms.
- DBSCAN [10]: By using appropriate density criteria, DBSCAN maintains the sparse background as noise.
- Approximate Rank-Order Clustering (ARO) [41]: ARO clusters data by an enhanced distance metric and approximative closest neighbor.
- Hierarchical Agglomerative Clustering (HAC) [42]: HAC uses a bottom-up approach, combines clusters, and stops clustering under predetermined rules.

- Spectral Clustering [12]: Spectral Clustering separates the data into several related components based on the similarity matrix between the data.
- Deep Density Clustering (DDC) [28]: DDC is the clustering of local neighborhoods based on their similarity in terms of density in the feature space.
- Consensus-driven Propagation Clustering (CDP) [31]: CDP is a learnable clustering algorithm that uses committee networks to improve robustness.
- L-GCN [2]: L-GCN is a learnable clustering technique that makes use of GCN to extract contextual data from the network for linkage prediction.
- Non-density division-GCN Clustering (NDD-GCN): A method that constructs an adaptive graph for all nodes as context without density division parts, then applies GCN for reasoning on it.
- Density Division Clustering based on GAE (DDC-GAE): A method that constructs an adaptive graph for all nodes as context with density division parts, then applies GAE for reasoning on it.

### 4.2. Experimental Results

4.2.1. Results and Performance Analysis

The comparative results between the DDC-GNN framework and other methods are shown in Table 4. The best results are highlighted in bold, while a "-" in the "Time" column indicates that the accuracy and time are not given in the original publications. We reproduce the experiments but lead to an "out of memory" result. The results demonstrate that our DDC-GNN framework outperforms other clustering approaches on IJB-B.

Since the predetermined number of clusters has a significant impact on K-Means performance, it might be challenging to achieve satisfactory results when the number of clusters is uncertain. Dealing with imbalanced data is out of the capability of spectral clustering and the data distribution has a significant impact on DBSCAN since it assumes that the density of various clusters is identical. Although HAC does not require a predetermined certain number of clusters, the iterative clustering process is time-consuming. The DDC-GNN framework performs better than ARO and DDC because the learnable connection is more accurate than unlearnable cosine similarity.

In the supervised methods comparison of the IJB-B datasets, the DDC-GNN framework outperforms other approaches. It is worth focusing that the results on IJB-B-512 are not the SOTA on the NMI metric, possibly because the amount of data is too small and the high-density and low-density parts cannot be separated well by tuning parameters $E$ and $M$. We also keep track of how long DDC-GAE/GCN and L-GCN take. L-GCN is less efficient than DDC-GAE and DDC-GNN because we have shortened the time required for a portion of the GNN subgraphs inference.

As seen in the results of MS-Celeb-1M datasets in Table 5, the DDC-GNN framework outperforms conventional clustering methods in MS-Celeb-1M. It outperforms L-GCN by 2%, which performs better than CDP, but slower than CDP since L-GCN is required to extract context information for building and inferring subgraphs. It can be seen that utilizing adaptive subgraphs as input can better forecast the connectivity of edges since the approach with non-density division (NDD-GCN) is superior to L-GCN. Calculations of L-GCN will take a lot of time because subgraphs are highly overlapped. However, compared with other supervised approaches, DDC-GAE with density division achieves a computation time reduction and DDC-GCN achieves a large calculation time reduction with comparable (or even slightly higher) performance. This demonstrates the accuracy and time-efficiency of the DDC-GNN framework.

**Table 4.** Comparison on IJB-B datasets.

| Method | IJB-B-512 | | | | | IJB-B-1024 | | | | | IJB-B-1845 | | | | |
|---|---|---|---|---|---|---|---|---|---|---|---|---|---|---|---|
| | F | NMI | ARI | SC(0.166) | Time | F | NMI | ARI | SC(0.170) | Time | F | NMI | ARI | SC(0.194) | Time |
| K-Means [9] | 0.639 | 0.856 | 0.447 | 0.181 | 175 s | 0.603 | 0.865 | 0.415 | 0.166 | 699 s | 0.600 | 0.868 | 0.340 | 0.164 | 2307 s |
| AP [40] | 0.494 | 0.854 | - | - | - | 0.484 | 0.864 | - | - | - | 0.477 | 0.869 | - | - | - |
| DBSCAN [10] | 0.768 | 0.904 | 0.776 | 0.187 | 11 s | 0.765 | 0.877 | 0.152 | 0.141 | 42 s | 0.737 | 0.851 | 0.065 | 0.116 | 150 s |
| Spectral Clustering [43] | 0.538 | 0.774 | 0.177 | 0.143 | 285 s | 0.522 | 0.780 | 0.126 | 0.133 | 1692 s | 0.551 | 0.840 | 0.296 | 0.136 | 3252 s |
| ARO [41] | 0.685 | 0.887 | 0.747 | 0.077 | 267 s | 0.599 | 0.880 | 0.634 | 0.063 | 551 s | 0.523 | 0.874 | 0.448 | 0.055 | 1062 s |
| SDCN [44] | 0.474 | 0.836 | - | - | - | 0.452 | 0.830 | - | - | - | - | - | - | - | - |
| DDC [28] | 0.802 | 0.921 | - | - | - | 0.805 | 0.926 | - | - | - | 0.800 | 0.929 | - | - | - |
| L-GCN [2] | 0.821 | 0.920 | 0.826 | 0.206 | 108 s | 0.819 | 0.928 | 0.846 | 0.110 | 198 s | 0.810 | 0.926 | 0.676 | 0.187 | 362 s |
| NDD-GCN | 0.826 | **0.926** | 0.830 | 0.209 | 97 s | 0.819 | 0.931 | 0.846 | 0.111 | 192 s | 0.812 | 0.930 | 0.676 | 0.181 | 350 s |
| DDC-GAE | 0.823 | 0.906 | 0.829 | 0.226 | 46 s | 0.817 | 0.933 | 0.829 | 0.108 | 70 s | 0.809 | 0.919 | 0.675 | 0.167 | 98 s |
| DDC-GCN | **0.829** | 0.917 | **0.833** | 0.207 | 32 s | **0.822** | **0.935** | **0.848** | 0.118 | 51 s | **0.816** | **0.931** | **0.677** | 0.175 | 75 s |

**Table 5.** Comparison on MS-Celeb-1M datasets.

| Methods | Precision | Recall | F-Score | NMI | Time |
|---|---|---|---|---|---|
| K-Means [9] | 52.52 | 70.45 | 60.18 | 94.56 | 13 h |
| DBSCAN [10] | 72.88 | 42.46 | 53.50 | 92.31 | **100 s** |
| HAC [42] | 66.84 | 70.01 | 68.39 | 92.92 | 18 h |
| ARO [41] | 81.10 | 7.30 | 13.34 | 84.68 | 250 s |
| CDP [31] | 80.19 | 70.47 | 75.01 | 94.69 | 350 s |
| L-GCN [2] | 84.40 | 72.36 | 78.86 | 94.76 | 2500 s |
| NDD-GCN | 87.24 | 74.98 | 80.65 | 95.47 | 2042 s |
| DDC-GAE | 82.76 | 79.96 | 80.85 | 95.49 | 650 s |
| DDC-GCN | **84.78** | **77.56** | **80.95** | **95.53** | 420 s |

### 4.2.2. Ablation Study

In the framework of DDC-GNN, we divide the data into high-density and low-density parts in the linked region division stage. Our experimental results will be directly impacted by how to form the high-density components of the data. Important factors for density division are $E$ and $M$. We also examine how $E$ and $M$ affect performance.

Because a bigger $E$ makes the estimated density of nodes more discriminative, Figure 5 illustrates that *F-score* increases as $E$ grows from 0.5 to 0.8, while *F-score* begins to decrease after 0.8. When $E$ is small, that means nodes might have few nodes around them and they are discriminative to each other. When $E$ is big, that means nodes might include many nodes from different clusters and this will affect the performance. Figure 6 illustrates that a greater $M$ also results in a higher *F-score* from 15 to 18. The *F-score* states to decrease after 18. This is because a too big M value makes DDC-GNN create big strong connected components in the high-density parts, which may result in wrong clusters. When M decreases, DDC-GNN generates small strong connected components, which rely on the last clustering step to link them back.

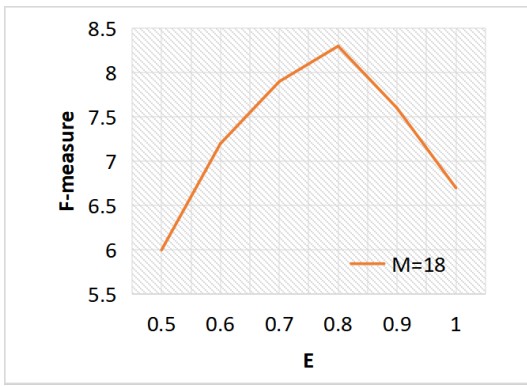

**Figure 5.** The impact of $E$.

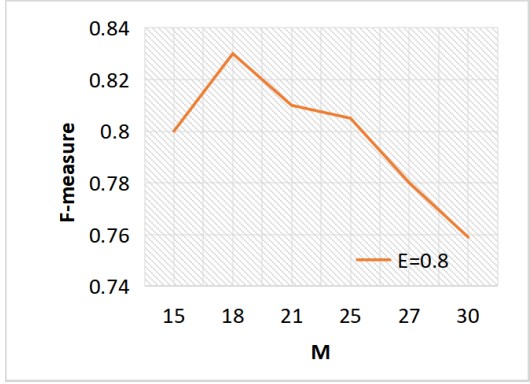

**Figure 6.** The impact of $M$.

When $E$ and $M$ rise to a certain amount, the performance achieves saturation. It is challenging to identify the nodes by density when distance $E$ is set too low since all densities tend to be similar tiny numbers. When $E$ is set too large, it is also easy to make the densities to be too smooth to be distinguished. For parameter $M$ in building high-density parts, a large $M$ will wrongly connect different cluster centers into one cluster while this type of inaccuracy occurs when setting a small $M$.

Considering the clustering efficiency and clustering accuracy, the values of $E$ and $M$ should not be too large, and when $E$ is 0.8 and $M$ is 18, the best performance can be obtained. As long as the data sources of face images are the same, the same $E$ and $M$ can be easily obtained, because of the same data distribution. While the parameter $K$ in K-means is a sensitive parameter to the clustering results and is hard to choose. Therefore, $E$ and $M$ are two insensitive parameters.

We implement experiments with various ratios of the high-density nodes/parts to the overall nodes. Figure 7 demonstrates that as $\rho$ grows, so does the clustering performance. The best performance achieves when $\rho$ is 0.4. The inference time becomes lower as $\rho$ rises, as seen in Figure 8. After a certain value, the increasing $\rho$ will result in a decrease in performance. This is because high-density nodes/parts are often located around the cluster center. If $\rho$ is too high, it will affect the accuracy of density division, which might result in wrong cluster central parts and worse performance.

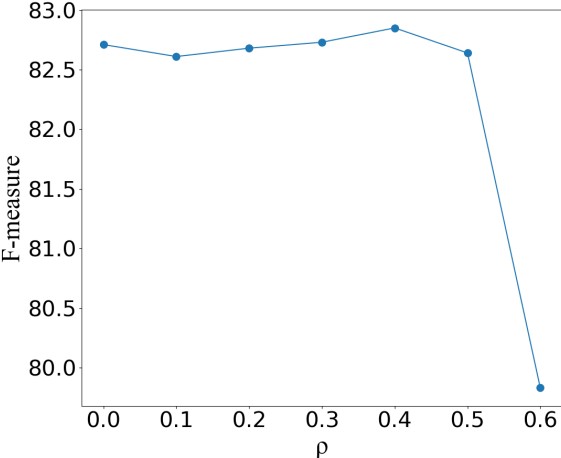

**Figure 7.** The impact of different high-density ratios on performance.

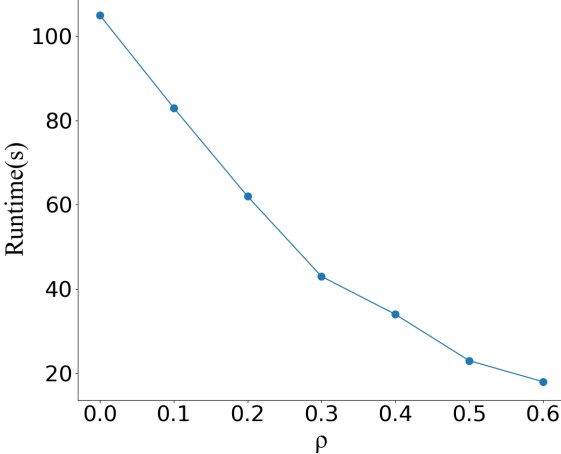

**Figure 8.** The impact of different high-density ratios on Runtimes.

The procedure of locating the nodes in the adaptive subgraph is influenced by two hyperparameters $k_1$ and $k_2$. We examine how different values for $k_1$ and $k_2$ affect the

adaptive subgraphs. We select a higher value $k_1 = 100$ since we anticipate that more supervised context information will be backpropagated during the training phase. At the same time, we set $k_2 = 10$ to avoid the adaptive subgraph being too big. In the testing step, we run a comparative test on IJB-B-512 to examine the effects of $k_1$ and $k_2$ on performance. Different $k_1$ and $k_2$ are employed in experiments and the results are shown in Figure 9.

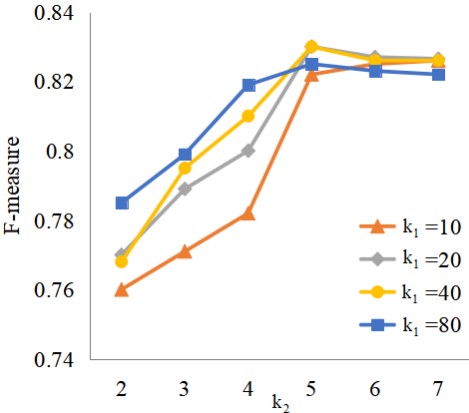

**Figure 9.** Sensitivity of performance with parameters $k_1$ and $k_2$.

It is obvious that the performance improves greatly with a bigger value of $k_1$ in the early stages. More additional neighbor nodes may be introduced by a bigger $k_1$, which will also increase the number of candidate node pairs (center node and its neighbors). As the value of $k_2$ increases, the overall performance shows an upward trend. The contextual information collected by the subgraph becomes increasingly adequate as $k_2$ increases in value. The performance achieves saturation at a specific level of $k_1$ and $k_2$. When $k_1$ and $k_2$ are in a specific range, it may be concluded that it can forecast the connectivity accurately.

## 5. Discussion

Existing face clustering methods based on graph neural networks leverage a large number of subgraphs to organize the data when clustering faces. These subgraphs are usually highly overlapping and incur redundant computational overheads which will lead to the inference speed being reduced. As the accuracy increases, more and more attention is focused on the time-efficiency.

We propose a more accurate face clustering method based on a density division framework with different connectivity predictions by graph convolutional neural networks, which extracts features from faces images and treats them as different nodes, and divides and builds the nodes into high-density parts and low-density parts based on local density values. The high-density parts have strong connectivity and form the core of different clusters. So we pay more attention to low-density parts. For each node in the low-density parts, an adaptive subgraph is generated based on the context, and the link prediction model and node classification model are used to reason about each subgraph to obtain a set of edge connectivities. The final connectivities predictions are obtained based on the weights of the edges connectivities to determine the connectivity possibility between the pair of nodes. The the high-density parts are merged into the low-density parts, and the pseudo-label propagation strategy is adopted to eliminate the unqualified edges by multiple iterations to obtain the final clustering.

Although the proposed graph neural network-based clustering method constructs adaptive subgraphs for nodes as input, it can contain more contextual information. The overall information is still local structural information, and it is difficult to grasp the structural information of the whole graph. Further research is needed to mine the global information to obtain better performance.

## 6. Conclusions and Future Work

We propose a supervised image clustering framework DDC-GNN based on density division and graph neural networks, which solves the problem of slow clustering speed caused by redundant subgraph calculation in the process of existing supervised face clustering. By a density division strategy that segments the face images into high-density and low-density parts, the DDC-GNN framework only infers the connectivities for images instance in the low-density parts, which improves the efficiency of face clustering significantly. In addition, DDC-GNN proposes constructing an adaptive subgraph in connectivity learning to capture the context information of a specific node. Experiments on MS-Celeb-1M and IJB-B show that the DDC-GNN framework is both time-efficient and effective.

Current clustering technology still needs a supervised training process and manual labeling. While manual labeling brings much effort when a large number of images are massive, clustering without manual labeling is a future research field, such as clustering by self-supervised learning or unsupervised learning.

**Author Contributions:** Q.Z. and L.L. wrote the manuscript and designed the comparative experiments; Y.C. supervised the study and revised the manuscript; Z.Y. gave suggestions to and revised the manuscript; Z.W. and W.S. contributed to the research design and manuscript writing. All authors have read and agreed to the published version of the manuscript.

**Funding:** This research is partially funded by the China National Key Laboratory Foundation of Underwater Measurement and Control Technology and Fundamental Research Funds for the Central Universities under Grant 3072022TS0601, and Singapore's National Research Foundation (NRF's) National Cybersecurity R&D Grant GC2018-NCR-0008.

**Data Availability Statement:** The links to public datasets are as follows, https://github.com/Zhongdao/gcn_clustering (accessed on 2 July 2022) and https://github.com/yl-1993/learn-to-cluster/blob/master/DATASET.md (accessed on 2 July 2022).

**Conflicts of Interest:** The authors declare no conflict of interest.

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
