# Peer review of "Efficient Supervised Image Clustering Based on Density Division and Graph Neural Networks"

_remotesensing, doi:10.3390/rs14153768_

Round 1
Reviewer 1 Report
The new version of the article definitely closes all the comments.
Reviewer 2 Report
I still have concerns over the reply, the authors gave, but nevertheless they added a study from 2020 which seems fine. I would like to recommend the acceptance of this work.
Reviewer 3 Report
From the second round review I have 3 concerns regarding this paper revision:
1. I suggested adding more metrics to measure the experiment results. The authors have done this.
2. About the difficulty of parameter tuning. The authors respond: "E and M can be easily obtained in the labeled training set. Moreover, they are not sensitive parameters". I only partly agree with these statements, as for real usage, we don't have labeled data for clustering.
3. About the performance for unbalanced data. The authors have done this.
As most of my concern has been addressed by the authors, I can accept this paper.
This manuscript is a resubmission of an earlier submission. The following is a list of the peer review reports and author responses from that submission.
Round 1
Reviewer 1 Report
The authors propose to use convolutional networks on graphs for image clustering. However, such a clustering problem is considered by them as a supervised learning problem. In this regard, comparison with K-means and DBScan (unsupervised learning) algorithms is not informative enough. Density separation is proposed to improve performance. This approach is probably justified in image segmentation. The work has been improved, however, the authors did not take into account some comments at all, which does not allow publishing the work at the moment.
Among these remarks, we highlight the following:
1) In the review part, segmentation algorithms based on pure convolutional networks should be considered (https://doi.org/10.3390/electronics11010130; https://doi.org/10.3390/s21082618; https://doi.org/10.1134/S1054661821030020 ) and model approach (K.-M. Lee and W. N. Street (1999) A fast and robust approach for automated segmentation of breast cancer nuclei. In: Proceedings of the IASTED International Conference on Computer Graphics and Imaging, 42–47; K. -M. Lee and W. N. Street (2000a) Automatic segmentation and classification using on-line shape learning, In: Proceedings of the 5th IEEE Workshop on the Application of Computer Vision, 64–70, https://doi.org/10.1134/ S105466181901005X; https://dollar.biz.uiowa.edu/~street/research/Mva061.pdf)
2) It would be interesting to compare the results with Mask R-CNN (https://arxiv.org/abs/1703.06870).
3) It would be interesting to add the results of processing individual images.
Reviewer 2 Report
The authors' response didn't fully address the concerns.
The authors said the accuracy of the first data set is almost equivalent to that of GCN-A, and the second dataset is obviously better.
In table 2, the results were not compared with GCN-A but GCN-M (Wang, Zhongdao, et al. "Linkage based face clustering via graph convolution network." Proceedings of the IEEE/CVF Conference on Computer Vision and Pattern Recognition. 2019.)
The result is obviously better than GCN-M, but not obviously better than GCN-A.
Reviewer 3 Report
The paper is presents a new clustering algorithm that is faster than the existing ones. Furthermore, the method has been used in conjunction with graph convolutional networks to showcase its efficacy. Although the idea is interesting, the presentation, comparative analysis, and use of English language seems weak.
English language problems. There are many spelling mistakes even in the abstract. Furthermore, there are long sentence, incoherency between paragraphs, and repetitive use of some specific words.
Presentation: In my opinion, the paper lacks proper structure, author need to improve the paper structure wise to improve the readability of this article.
comparative analysis: The article is mainly about the clustering algorithm however, the comparative analysis is mostly performed against old methods. For instance, in table 2, the comparative analysis is performed with only one method that is from 2018 and the others are way older. In table 3, the most recent method is from 2019. Authors should compare with atleast five to six more works from 2019-2021 (2022).
time comparison: It is not explicitly mentioned in the manuscript if authors have reproduced all the methods that are being compared, if not? the comparison is not justified. The authors should also take into account this condition when comparing with other recent works as suggested in previous comment. A mode of fair comparison for time and space complexity should be adopted in order to justify the claim that has been made in the manuscript.
Reviewer 4 Report
A few notes from the previous round that have not been addressed:
- Along with NMI, the most widely used Clustering metrics are the Adjusted Rand Index and Silhouette Coefficient. Please use these metrics to improve this paper.
- The authors' approach although efficient also suffers a similar problem as K-Means, in this case determining E and M parameters rather than the number of clusters in K-Means.
- One of the most difficult problems in clustering is to cluster unbalanced data. The author should add results on clustering unbalanced face data using their approach and compare the results.
Round 2
Reviewer 1 Report
In general, the article has been improved.
However, I cannot fully agree that all the articles not included in the review are about segmentation based on labeled data. For example, in the article https://doi.org/10.1134/S105466181901005X there is no mention of pre-markup, and I once again recommend the authors to pay attention to it.
Reviewer 3 Report
I am not satisfied with the responses and justification given by authors. There are several ways to comply with the method, either authors have provided comparative analysis on a mutual dataset to show the efficacy of their method or reproduced the results from studies having publicly available codes. Having said that, authors mentioned that image clustering field is not very popular yet there are several studies published in 2021 alone. Just to mention a few:
"Deep Face Clustering using residual graph convolutional network"
"Learning Deep Discriminative Representations with Pseudo Supervision for image clustering"
"Improving Unsupervised Image Clustering with Robust Learning"
"Graph Regularized Residual Subspace Clustering Network for Hyperspectral Image Clustering"
"Image Deep Clustering based on Local-Topology Embedding"
"Deep Self-Representative Subspace Clustering Network"